# Unraveling the Roles of Epigenetic Regulators During the Embryonic Development of *Rhipicephalus microplus*

**DOI:** 10.3390/ijms26189171

**Published:** 2025-09-19

**Authors:** Anderson Mendonça Amarante, Daniel Martins de Oliveira, Marcos Paulo Nicolich Camargo de Souza, Manoel Fonseca-Oliveira, Antonio Galina, Serena Rosignoli, Angélica Fernandes Arcanjo, Bruno Moraes, Alessandro Paiardini, Dante Rotili, Juan Diego de Paula Li Yasumura, Sarah Henaut-Jacobs, Thiago Motta Venancio, Marcelle Uhl, Rodrigo Nunes-da-Fonseca, Luis Fernando Parizi, Itabajara da Silva Vaz Junior, Claudia dos Santos Mermelstein, Thamara Rios, Lucas Tirloni, Carlos Logullo, Marcelo Rosado Fantappié

**Affiliations:** 1Instituto de Bioquímica Médica Leopoldo de Meis, Centro de Ciências da Saúde, Universidade Federal do Rio de Janeiro (UFRJ), Rio de Janeiro 21941-599, Brazil; amarante@bioqmed.ufrj.br (A.M.A.); daniel-thomasi@hotmail.com (D.M.d.O.); mnpaulog20191@gmail.com (M.P.N.C.d.S.); manoel.oliveira@bioqmed.ufrj.br (M.F.-O.); galina@bioqmed.ufrj.br (A.G.); angelicaarcanjo@hotmail.com (A.F.A.); bmarques@bioqmed.ufrj.br (B.M.); juandiegoyasumura@yahoo.com.br (J.D.d.P.L.Y.); thamara.rios@bioqmed.ufrj.br (T.R.); 2Instituto Nacional de Entomologia Molecular, Universidade Federal do Rio de Janeiro (UFRJ), Rio de Janeiro 21941-599, Brazil; 3Centre for Regenerative Medicine “Stefano Ferrari”, Department of Life Sciences, University of Modena and Reggio Emilia, 41125 Modena, Italy; serena.rosignoli@unimore.it; 4Department of Biochemical Sciences “A. Rossi Fanelli”, Sapienza University of Rome, 00185 Rome, Italy; alessandro.paiardini@uniroma1.it; 5Department of Science, “Roma Tre” University, Viale Guglielmo Marconi 446, 00146 Rome, Italy; dante.rotili@uniroma3.it; 6Biostructures and Biosystems National Institute (INBB), Via dei Carpegna 19, 00165 Rome, Italy; 7Laboratório de Química e Função de Proteínas e Peptídeos, Centro de Biociências e Biotecnologia, Universidade Estadual do Norte Fluminense Darcy Ribeiro, Campos dos Goytacazes, Rio de Janeiro 28015-622, Brazil; henautjacobs@gmail.com (S.H.-J.); thiago.venancio@gmail.com (T.M.V.); 8Instituto de Biodiversidade e Sustentabilidade-NUPEM, Universidade Federal do Rio de Janeiro (UFRJ), Av. São José do Barreto 764, Macaé 27965-550, Brazil; marcelle_uhl@yahoo.com.br (M.U.); rodrigo.nunes.da.fonseca@gmail.com (R.N.-d.-F.); 9Centro de Biotecnologia e Faculdade de Veterinária, Universidade Federal do Rio Grande do Sul (UFRGS), Porto Alegre 91509-900, Brazil; luisfparizi@gmail.com (L.F.P.); itabajara.vaz@ufrgs.br (I.d.S.V.J.); 10Instituto de Ciências Biomédicas, Universidade Federal do Rio de Janeiro (UFRJ), Rio de Janeiro 21941-590, Brazil; mermelstein@histo.ufrj.br; 11Tick-Pathogen Transmission Unit, Laboratory of Bacteriology, Division of Intramural Research, National Institute of Allergy and Infectious Diseases, Hamilton, MT 59840, USA; lucas.tirloni@nih.gov

**Keywords:** *Rhipicephalus microplus*, epigenetics, mitochondrial function

## Abstract

Epigenetic modifications are long-lasting changes to the genome that influence a cell’s transcriptional potential, thereby altering its function. These modifications can trigger adaptive responses that impact protein expression and various cellular processes, including differentiation and growth. The primary epigenetic mechanisms identified to date include DNA and RNA methylation, histone modifications, and microRNA-mediated regulation of gene expression. The intricate crosstalk among these mechanisms makes epigenetics a compelling field for the development of novel control strategies, particularly through the use of epigenetic drugs targeting arthropod vectors such as ticks. In this study, we identified the *Rhipicephalus microplus* orthologs of canonical histone-modifying enzymes, along with components of the machinery responsible for m^5^C and ^6^mA-DNA, and m^6^A-RNA methylations. We further characterized their transcriptional profiles and enzymatic activities during embryonic development. To explore the functional consequences of epigenetic regulation in *R. microplus*, we evaluated the effects of various epigenetic inhibitors on the BME26 tick embryonic cell line. Molecular docking simulations were performed to predict the binding modes of these inhibitors to tick enzymes, followed by in vitro assessment of their effects on cell viability and morphology. Tick cells exposed to these inhibitors presented phenotypic and molecular alterations. Notably, we observed high levels of DNA methylation in the nuclear genome. Importantly, inhibition of DNA methylation using 5′-azacytidine (5′-AZA) was associated with increased activity of the mitochondrial electron transport chain and ATP synthesis but reduced cellular proliferation. Our findings highlight the importance of epigenetic regulation during tick embryogenesis and suggest that targeting these pathways may constitute a novel and promising strategy for tick control.

## 1. Introduction

The tick *R. microplus* is an obligatory hematophagous ectoparasite that causes major losses to bovine herds. *R. microplus* is a one-host tick, and its life cycle consists of the free-living and the parasitic phases [1]. Ticks require a blood meal at each stage (larva, nymph, and adult) to develop into the next stage. Tick eggs do not require external nutrients and are reliant on the yolk provided within the egg for sustenance during development [2].

The economic losses associated with *R. microplus* parasitism are due to direct effects of the tick itself, which causes skin injuries and long-standing blood loss, leading to anemia and reductions in both weight gain and milk production, or are produced indirectly via transmission of tick-borne pathogens such as *Babesia* spp. and *Anaplasma marginalev* [1,3]. Despite the considerable impact of the ticks on the economy and human health, current tick control strategies still rely mostly on the use of chemical acaricides. This is recognized as a worldwide drawback to successful tick control. Although the immunization of cattle against *R. microplus* and other ticks has been recognized as an alternative approach against chemical control strategies, up to this date, an efficient and/or viable vaccine is still lacking [4,5,6]. Therefore, a deeper understanding of tick physiology is needed to identify new molecular targets that can be useful in the development of novel tick control methods.

Epigenetic regulation has been proposed in humans as a cancer therapy [7]. A promising strategy for controlling gene expression in eukaryotes that can cause harm to humans and animals has been hypothesized [8,9,10]. Therefore, targeting gene regulatory mechanisms such as epigenetics would contribute immensely to controlling or even eradicating vectors that strongly impact the economy, as well as on human and animal health.

Epigenetic information directs the formation of distinct cellular and organismal phenotypes from a common genome. For example, the ability of insects to develop phenotypes appropriate to their environment relies on epigenetic information [11,12,13,14,15]. Epigenetics is specifically concerned with heritable changes in gene regulation that occur in response to intercellular and extracellular environmental cues. Broadly defined, epigenetic information can take many forms, as factors at many levels can stably affect gene regulation. Thus, the field of molecular epigenetics is generally concerned with molecular mechanisms that directly affect, alter, or interact with chromatin.

In arthropods as well as mammals, epigenetic modifications play significant roles in the control of multiple biological processes that regulate development, growth, reproduction, behavior, immunity, and castle differentiation [16,17,18,19,20,21].

DNA methylation is perhaps the most faithfully heritable form of epigenetic information. In recent years, DNA methylation has nevertheless captured the attention of insect and tick researchers, driven in large part by a desire to understand the importance of DNA methylation to developmental plasticity [22,23,24,25,26].

The majority of DNA in the metazoan nucleus is incorporated into nucleosomes, and many regulatory processes in eukaryotes have been linked to alterations of histone–DNA interactions [27,28]. There are several important ways in which nucleosomes can be altered to impact the regulation of genes. Histone posttranslational modifications (hPTMs) are a diverse set of epigenetic signals that typically occur on the N-terminal amino acid tails of histone proteins [27,28]. There are several ways in which hPTMs can alter transcription. First, the association between the target histone and underlying DNA can be directly impacted by the addition of an acetyl or methyl groups to a histone protein, which may consequently increase or decrease the ability of transcription factors to access DNA [27,28].

Histone proteins have been studied extensively in *Drosophila* [17,29], but only recently have histones been investigated in nonmodel insect taxa [30,31,32]. Histone protein modifications are functionally conserved in essentially all eukaryotes, including the arthropods [30,31,32]. Thus, in arthropods, histone proteins may be directly involved in the mediation of phenotypic plasticity [33].

N^6-^methyladenosine (m^6^A) is the most abundant internal epigenetic modification in eukaryotic mRNAs and has profound effects on RNA metabolism, including RNA stability [34], splicing [35], translation [36] and RNA-protein interactions [37]. In the last decade, numerous studies have revealed that m^6^A modification plays a vital role in regulating eukaryotic growth and development [38], cell differentiation [39], reproduction [40], the DNA damage response [41], circadian rhythms [42], and cancer induction [43].

The importance of epigenetic modifications during mammalian embryogenesis has been well described and characterized [44]. However, the role of epigenetics in arthropod embryogenesis, with the exception of *Drosophila*, is far from understood [45,46,47]. In this context, the roles of epigenetics in the biology of an important blood-feeding ectoparasite group of arthropods, the ticks, are still poorly understood [22,48,49] and, thus, deserve attention.

In the present work, we aimed to characterize the roles of epigenetic modifications in *R. microplus* during embryogenesis, thus providing a new layer of knowledge for tick biology.

## 2. Results

### 2.1. The Epigenetic Machinery Is Encoded Within the R. microplus Genome

Given the limited number of epigenetic studies in ticks and the vast diversity of histone modifications in eukaryotic chromatin, we chose to investigate the presence of some (but not all) epigenetic enzyme homologs in the *R. microplus* genome (Figure 1 and Appendix A). For histone-modifying enzymes, we focused on two major histone acetyltransferases (HATs), CBP/p300 and GCN5, along with six histone methyltransferases (HMTs): EZH2, EHMT, SETDB, SETD2, SETD4, and SETD7. Additionally, we examined three histone deacetylases, HDAC1, HDAC4, and HDAC8, which represent Class I, II, and III, respectively. We also identified the complete set of enzymes responsible for 5-methylcytosine (m^5^C) DNA methylation, including DNMT1, DNMT3A, and DNMT3B, as well as components involved in N6-methyladenosine (m^6^A) RNA methylation: the writer proteins METTL3, METTL14, and WTAP, and the reader proteins YTHDC and YTHDF. Importantly, the roles of these epigenetic enzymes in embryonic development of other organisms have been well documented [50,51,52,53,54,55,56,57].

Analysis of the sequence alignments for all proteins (Appendix A), along with the percentage identity data (Appendix A), revealed high levels of conservation within the catalytic domains between tick and mammalian homologs.

Phylogenetic and domain architecture analyses of *R. microplus* Ezh2 orthologs in related arthropods revealed distinct evolutionary patterns and conserved functional domains (Appendix A). OrthoFinder identified 45 orthologs across 20 species, including *Apis*, *Solenopsis*, and *Drosophila*. Phylogenetic reconstruction with IQ-TREE2 supported strong clustering of *R. microplus* with *Ixodes scapularis* (XP_029824349.2, XP_029824350.2), reflecting their shared taxonomic order (Parasitiformes), with bootstrap values >90%. Domain annotation via HMMER/Pfam confirmed that conserved functional motifs namely, the SET, preSET_CXC, PRC2_HTH_1, and EZH2_MCSS domains were universally present, whereas the FLYWCH and Vfa1 domains were lineage specific.

Phylogenetic and domain architecture analyses of *R. microplus* CBP/p300 orthologs (Appendix A) revealed conserved functional domains and evolutionary relationships. Phylogenetic reconstruction using IQ-TREE2 demonstrated strong clustering of *R. microplus* with arachnid species such as *Parasteatoda tepidariorum* (XP_015910461.1, XP_019822458.1) (bootstrap > 85%), reflecting shared chelicerate ancestry. Domain annotation using HMMER/Pfam confirmed the universal presence of most domains, with only KIX and KIX_2 being alternated in certain taxa. Multiple sequence alignment (MAFFT) highlighted divergent regions in the Creb-binding domain of *R. microplus* and its cluster, suggesting functional divergence unique to acarines.

### 2.2. Embryonic Development of R. microplus

To investigate epigenetic dynamics during *R. microplus* embryogenesis, we reproduced a previously established protocol that defines embryological stages using DAPI staining [58], covering the developmental window from Days 6 to day 21 (Figure 2, showing both lateral and dorsal views). Although the initial cleavages occur between Days 1 and 3 marked by early cell division and migration toward the yolk periphery, we chose not to include these stages in our study. This decision was based on the low number of cells and nuclei present at these time points, as indicated by the minimal detection of histone molecules (Appendix A). The end of days 5 and 6 corresponds to stage 7 of development [58], during which the germ band is visible along with defined head, thoracic segments, and a posterior growth zone. Segmental grooves begin to emerge at this stage. By day 9 (stage 10), the first three pairs of legs (L1–L3) elongate and reach the ventral midline, while the fourth leg (L4) remains undeveloped. Leg segmentation becomes evident, and the germ band begins to retract. On Day 12 (stage 11), the posterior region continues retracting and migrates ventrally in an inverted orientation. The legs shift to a lateral position, and the embryo begins to appear more condensed. Leg segmentation becomes more pronounced in L1–L3, whereas L4 remains minimal. The chelicerae and pedipalps also relocate to the ventral side. Between Days 15 and 18 (stage 13), dorsal closure nears completion, and the prosoma reaches its final anatomical position. From days 18 to 21 (stage 14), the larva is fully formed and prepares to hatch from the egg.

### 2.3. Transcriptional Profile of Epigenetic Proteins Throughout R. microplus Embryonic Development

We assessed the transcript levels of genes encoding enzymes involved in histone, DNA, and RNA modifications in *R. microplus* embryos (Figure 3A–C). As shown in Figure 3A, the expression levels of the two histone acetyltransferases (HATs), CBP/p300 showed elevated expression from Day 6 to day 15, followed by a marked decrease between Days 18 and 21. In contrast, Gcn5 expression increased during the later stages of development, from Day 15 to 21. Among the histone methyltransferases (HMTs), EZH2, EHMT, and SETD4 maintained relatively constant expression throughout embryogenesis (Days 6–21). In contrast, SETDB and SETD2 presented relatively high transcript levels during early development (Days 6–15), whereas SETD7 presented a distinct pattern: elevated expression from Day 6 to 12, a nearly complete absence of transcripts from Day 15 to 18, and a significant reappearance of transcripts on Day 21.

For the DNA methylation enzymes (Figure 3B), DNMT1 and DNMT3a expression appeared to be restricted to the early stages of development (days 6 to 12), with DNMT1 showing notably lower transcript levels compared to DNMT3a. In contrast, DNMT3b exhibited high expression levels throughout the entire developmental period.

Regarding the m^6^A RNA methylation machinery (Figure 3C), the genes encoding the two writer enzymes, METTL3 and METTL14, exhibited relatively constant expression throughout embryogenesis (Days 6–21), with METTL14 showing significantly higher transcript levels than METTL3. Among the genes encoding m^6^A reader proteins, YTHDC transcription began on Day 12 and remained stable through Day 21. In contrast, YTHDF was expressed at very low levels at Days 6 and 9, was undetectable on Day 12, and reappeared from Days 15 to 21.

### 2.4. Activity of the Epigenetic Enzymes Throughout R. microplus Embryonic Development

We observed significant modulation of histone modifications during tick embryogenesis (Figure 4A–F). Days 1 and 3 of embryogenesis were not included in our analysis because only minimal detection of histone molecules was observed at these time points (as shown in Appendix A). We examined three histone marks associated with gene activation: H3K27ac, catalyzed by CBP/p300; H3K14ac, catalyzed by Gcn5; and H3K4me3, catalyzed by SETD2, SETD4, and/or SETD7 (Figure 4A–C). CBP/p300 activity was nearly undetectable on Day 6 of embryonic development but peaked on Days 9 and 12, followed by a gradual decline from Days 15 to day 21 (Figure 4A). In contrast, Gcn5 displayed an inverse pattern, with minimal acetylation on Days 6 and 9, and a progressive increase in activity from Day 12 through Day 21 (Figure 4B). Similarly, the activity of SETD2/4/7 was weak on Day 6 but increased markedly from Days 9 to day 21 (Figure 4C). With respect to repressive histone marks, the activities of EZH2, SETDB, and EHMT were undetectable on Days 6 and 9, but increased significantly from Days 12 to day 21 (Figure 4D–F). Notably, EzH2 and EHMT Exhibited particularly high activity during the later stages of embryogenesis (Figure 4D,F).

With respect to nucleic acid modifications, significant changes were observed only in mRNA m^6^A methylation (Figure 5A). Specifically, METTL3/METTL14 activity was high on Days 6 and 9, followed by a marked and gradual decline from day 12 to day 21 (Figure 5A). In contrast, the ^6^mA and m^5^C DNA methylation levels remained relatively stable throughout embryonic development (Figure 5B,C, respectively).

As shown in Figure 5B, we conducted control experiments to rule out RNA contamination in our genomic DNA (gDNA) samples (Appendix A). As shown in Appendix A, a small amount of RNA contamination was detected in gDNA extracted from BME26 cells or eggs (arrows). This contamination was effectively eliminated by RNase A treatment (Appendix A, Lanes 3 and 6). The purified gDNA samples presented high levels of ^6^mA methylation (Appendix A).

Similarly, as shown in Appendix A, control experiments were performed to confirm that the detected m^5^C methylation (Figure 5C) was specific to gDNA. In this case, no m^5^C methylation was detected in RNA (RNA was present, as shown in Appendix A) from either BME26 cells or eggs (Appendix A).

### 2.5. Biological Effects of Epigenetic Inhibitors on BME Cells

BME26 cells were tested with three epigenetic inhibitors (Figure 6A–C,E,H) that have been extensively validated in mammalian systems: STM2457, a synthetic inhibitor of METTL3 [59]; 5′-azacytidine (5′-AZA), an inhibitor of DNA methyltransferases (DNMTs) [51]; and trichostatin A (TSA), an inhibitor of histone deacetylases (HDACs) [60].

Prior to treatment, we performed molecular docking simulations to evaluate the potential binding of these compounds to their respective tick homologs (Figure 6A–H). Specifically, TSA was docked with HDAC1, HDAC4, and HDAC6, representing Class I, Class IIa, and Class IIb HDACs, respectively (Figure 6A–C); STM2457 was docked with the METTL3/METLL14 heterodimer (Figure 6E), and 5′-AZA was docked with DNMT1 (Figure 6G). The docking simulations revealed that all compounds engage their tick enzyme targets with very favorable binding energies and in virtually identical poses to those observed in the corresponding human complexes (Figure 6A–C,E,G). TSA docks into the catalytic tunnel of tick HDAC1, HDAC4, and HDAC6 with calculated binding free energies of −10.1 kcal/mol, −9.1 kcal/mol, and −10.5 kcal/mol, respectively. In each case, the hydroxamate zinc-chelating group of TSA coordinates the active-site Zn^2+^ ion in a bidentate fashion, whereas the cap group forms a network of hydrogen bonds to conserved residues (e.g., Y306 and H141 in HDAC1). A superposition of the tick and human HDAC6•TSA complex (PDB: 5EDU) shows an RMSD of 0.6 Å over all heavy atoms and an identical arrangement of the aromatic “cap” ring against the enzyme surface. Similarly, 5′-AZA occupies the DNMT1 catalytic pocket with a binding energy of −8.7 kcal/mol. The nucleoside analog establishes two key hydrogen bonds between its ring nitrogen and Glu128, and a covalent bond with the side chain of Cys88. The tick-DNMT1•5′-AZA and human-DNMT1•5′-AZA structures overlay with an RMSD of less than 1 Å and retain precisely the same hydrogen-bonding pattern. Finally, STM2457 binds the methyltransferase heterodimer METTL3–METTL14 with highly favorable energies (−9.3 kcal/mol). Its adenosine-mimetic scaffold slots into the SAM-binding cleft, forming hydrogen bonds to Asp13 and Asn185 of METTL3, whereas the imidazo [1,2-a]pyridin-2-ylmethyl] ring tail engages in a conserved stacking interaction with Arg172. Finally, Asp31 and Ser147 interact with the N6-of the [(cyclohexylmethylamino)methyl] moiety. Overlaying the tick and human METTL3 structures confirmed an identical binding mode, with both complexes displaying superimposable ligand orientations and contacting residues (RMSD ≈ 0.8 Å). Together, these in silico dockings suggest not only that TSA, 5′-AZA, and STM2457 bind strongly to their tick enzyme homologs, but also that their molecular recognition profiles are essentially indistinguishable from those of human targets.

To assess whether the inhibitors exert cytotoxic effects on BME26 cells, we conducted a cell viability assay based on ATP quantification (Figure 6D,F,H). Treatment with either TSA or STM2457 resulted in a significant reduction in ATP levels at concentrations of 50 and 100 μM (Figure 6D,F), indicating decreased cell viability. In contrast, cells treated with 5′-AZA presented increased ATP production across the 25 to 100 μM concentration range (Figure 6H), suggesting a potential increase in cellular metabolic activity, such as increased mitochondrial function or proliferation. However, the possibility of glycolytic pathway overactivation cannot be ruled out.

To gain more detailed and precise insights into the cytotoxic effects of the inhibitors on cell morphology, we performed confocal microscopy analyses. A clear and significant in histone acetylation (H3K27ac) was observed in BME26 cells treated with TSA for 48 h (Appendix A). We further confirmed histone hyperacetylation in TSA-treated cells by Western blot analysis (Appendix A). Notably, the Western blot analysis also revealed insights into the crosstalk between H3K27ac and H3K27me3 in BME26 cells, with the acetylation mark (H3K27ac) being overridden by the repressive methylation mark (H3K27me3) (Appendix A).

BME26 cells were then treated with STM2457, and their actin cytoskeleton was analyzed using phalloidin staining (Figure 7). The inhibition of RNA-m^6^A methylation in BME26 cells resulted in alterations in actin filaments (Figure 7A,B, arrows). STM2457-treated cells presented an increase in the number of actin filaments (F-actin) suggesting that RNA-m^6^A methylation might be involved in the regulation of actin polymerization in BME26 tick cells. In the DMSO-treated cells most of the phalloidin staining was concentrated in small aggregates of F-actin, possibly containing short actin filaments.

### 2.6. m^5^C Methylation of the Nuclear Genome and Electron Transport Chain Function in BME Cells

Following the unexpected observation of elevated intracellular ATP levels after inhibiting DNA methylation with 5′-AZA (Figure 6H), we proposed that this metabolic change might be connected to alterations in mitochondrial epigenetic regulation. Another possibility was that nuclear DNA methylation could exert an indirect regulatory influence over mitochondrial function. To investigate these hypotheses, we performed confocal microscopy on BME26 tick embryonic cells subjected to 5′-AZA treatment. Our imaging analysis revealed that under normal conditions, these cells exhibited robust nuclear DNA methylation signals, detected through m^5^C (5-methylcytosine) staining. However, upon treatment with 5′-AZA, there was a pronounced decline in m^5^C signal intensity within the nucleus (Figure 8A). This finding strongly suggests that in ticks, DNA methylation activity is confined to the nuclear genome, with no detectable methylation occurring within the mitochondrial DNA.

Furthermore, over the course of six days posttreatment with 5′-AZA, we noted a progressive and statistically significant decrease in the total number of viable cells (Figure 8B). Intriguingly, this reduction in cell number was consistently accompanied by a sustained and significant increase in ATP levels (Figure 8C). These findings point toward a potential link between the nuclear DNA methylation status and bioenergetic homeostasis in BME26 cells. To confirm this, we performed respirometry experiments and clearly demonstrated that cellular respiration rates were significantly elevated in cells treated with 5′-AZA, as shown in Figure 8D,E. This increase in oxygen consumption strongly suggests an overactivation of the mitochondrial electron transport chain, implying that nuclear DNA demethylation induced by 5′-AZA may enhance mitochondrial metabolic activity.

## 3. Discussion

This study provides a detailed characterization of the epigenetic machinery involved in *R. microplus* embryogenesis and offers new insights into how epigenetic regulation may influence tick development and physiology. Our findings significantly expand the current understanding of *R. microplus* epigenetics [22,48,49], a relatively underexplored field in vector biology.

We identified a set of conserved epigenetic enzymes in the *R. microplus* genome, including key histone acetyltransferases (HATs), histone methyltransferases (HMTs), histone deacetylases (HDACs), and the enzymatic machinery required for DNA (m^5^C and ^6^mA) and RNA (m^6^A) methylation. High sequence similarities within catalytic domains between tick and mammalian orthologs imply functional conservation and provide a strong rationale for investigating small-molecule inhibitors validated in other systems.

By mapping transcriptional profiles and enzymatic activities during embryonic development, we demonstrated that the expression and function of epigenetic regulators in *R. microplus* are temporally dynamic and stage specific. For example, the acetyltransferase CBP/p300 showed peak activity during mid-embryogenesis (days 9–12), corresponding to periods of rapid tissue differentiation, whereas Gcn5 activity increased during late embryogenesis (Days 15–21), potentially supporting terminal maturation processes. In contrast, several repressive HMTs (e.g., EZH2 and EHMT) displayed increased activity only during the later stages, suggesting roles in developmental gene silencing and chromatin condensation.

EZH2 (enhancer of Zeste homolog 2), a core component of the polycomb repressive complex 2 (PRC2), is a histone methyltransferase enzyme that plays a crucial role in gene silencing by adding methyl groups to histone H3 at lysine 27 [52,53]. EzH2 is essential for early embryonic development, particularly for cell lineage determination, embryonic stem cell (ESC) maintenance, and proper differentiation of trophoblast cells [52,53,54]. In Drosophila, Polycomb group (PcG) proteins are crucial for maintaining the silenced state of genes, particularly homeotic genes, which determine body segment identity [52].

Histone H3 lysine 27 (H3K27) is an antagonistic switch between PcG-mediated repression and activation by trithorax group (TrxG) proteins. TrxGs counteract PcGs through the H3K27 demethylase UTX and recruit the histone acetyltransferase (HAT) CREB-binding protein (CBP) and its paralog p300 to acetylate H3K27 (H3K27ac) and mutually exclude H3K27me3 [54]. Importantly, previous studies have shown that a global loss of H3K27 methylation leads to aberrant accumulation of H3K27 acetylation, driven by the histone acetyltransferases CBP and p300 [53]. This hyperacetylation occurs at genomic regions where H3K27 methylation is normally present, including Polycomb group (PcG)-bound promoters and non-lineage-specific enhancer elements [54].

Our findings that the dynamic interplay between H3K27 acetylation (H3K27ac) and trimethylation (H3K27me3) also occurs in *R. microplus* (compare Figure 4A,D), suggest that a similar epigenetic regulatory mechanism is conserved. Given the importance of these modifications in controlling gene expression during development, this interplay could serve as a novel target for strategies aimed at disrupting tick development and reproduction, providing a potential avenue for controlling this economically important ectoparasite.

In *R. microplus*, we observed distinct expression patterns among the DNA methyltransferases. DNMT1 was expressed primarily during early embryogenesis (Days 6–12), which is consistent with its role in de novo DNA methylation [55]. Interestingly, DNMT3a and DNMT3b remained highly expressed and active in essentially all stages, suggesting a sustained role in embryonic DNA methylation dynamics in ticks, potentially also including the regulation of repetitive elements or noncoding regions [61].

Previous research has shown that ^6^mA is highly dynamic during early embryogenesis in *Drosophila*. Interestingly, the temporal pattern of ^6^mA dynamics closely coincides with the maternal-to-zygotic transition (MZT) [62,63,64]. The Forkhead box (Fox) family protein Jumu acts as a maternal transcription factor by preferentially binding to ^6^mA-marked DNA, thereby regulating embryonic gene expression and contributing to MZT, at least in part through the regulation of Zelda [64].

In contrast, our results demonstrated that ^6^mA DNA methylation levels remain constant throughout *R. microplus* embryogenesis, suggesting that this epigenetic modification may play a different role during tick embryonic development.

In *R. microplus*, the m^6^A RNA methylation machinery, specifically the methyltransferase complex components METTL3 and METTL14, was found to be expressed across all developmental stages. However, their enzymatic activity peaked during early embryogenesis, indicating a dynamic shift in post-transcriptional gene regulation during this critical period. Notably, the highest levels of METTL3/14 activity were observed as early as six days postoviposition, a timeframe that coincides with major developmental milestones. This temporal pattern suggests that m^6^A RNA methylation may play a pivotal role in orchestrating the maternal-to-zygotic transition (MZT), a tightly regulated process in which the control of gene expression shifts from maternally deposited transcripts to zygotic genome activation. The enrichment of METTL3/14 activity at this stage implies that m^6^A modifications may be essential for regulating transcript stability, splicing, translation efficiency, or degradation during MZT, thereby contributing to proper embryonic development in this tick species. In fact, deletion of the m^6^A-binding protein YTHDF2 in zebrafish embryos leads to reduction in the decay of m^6^A-modified maternal mRNAs and the suppression of the zygotic genome activation. Upon deletion of YTHDF2, the embryos fail to initiate timely MZT, undergo a cell-cycle pause, and remain developmentally delayed throughout larval life [65]. In addition, studies have shown that m^6^A methylation regulates the embryonic development of insects by affecting maternal mRNA decay, allowing normal embryogenesis in *Drosophila* [66].

Pharmacological inhibition of epigenetic regulators in BME26 cells corroborated their functional importance. Histone deacetylase inhibition by TSA led to histone hyperacetylation and a reduction in ATP levels, which was consistent with disrupted chromatin organization, and decreased cell viability. These effects were mirrored by inhibition of m^6^A-RNA methylation, which also reduced viability and caused visible cytoskeletal disruptions (Figure 7).

Although not yet directly shown in ticks, m^6^A RNA methylation could modulate actin filament dynamics indirectly by controlling the expression of mRNAs encoding actin regulators [67]. This phenomenon is especially relevant during early development, feeding, molting, and immune responses. In this context, key indirect m^6^A-regulated pathways are emerging based on what is known in arthropods and other invertebrates [68].

One of the most striking discoveries in this study is the observation of high levels of 5-methylcytosine (m^5^C) DNA methylation within the nuclear genome, accompanied by unexpectedly strong effects on mitochondrial physiology. Specifically, this nuclear methylation pattern is associated with enhanced electron transport chain activity and elevated ATP synthesis (Figure 8). Such a link between nuclear epigenetic status and mitochondrial output highlights a previously underappreciated axis of metabolic regulation. Although the precise molecular mechanism underlying this crosstalk remains unresolved, it is reasonable to hypothesize that in ticks, nuclear DNA methylation modulates the transcription of genes encoding mitochondrial proteins. This would include structural components of the electron transport chain, key metabolic enzymes involved in oxidative phosphorylation and substrate utilization, as well as regulatory factors for mitochondrial biogenesis, for example the peroxisome proliferator-activated receptor gamma coactivator 1-alpha (PGC-1α) and mitochondrial transcription factor A (TFAM). Under normal conditions, methylation of promoter regions in these nuclear-encoded genes might repress their transcription. Inhibition of DNA methylation, as in the presence of 5′-AZA, could lead to gene repression, enabling increased transcription of mitochondrial biogenesis regulators and components of the electron transport chain. This transcriptional upregulation could, in turn, elevate mitochondrial respiration capacity, resulting in greater ATP yield.

Such a mechanism would position nuclear DNA methylation not only as a regulator of genomic expression but also as a pivotal upstream modulator of cellular energy metabolism. If confirmed, this finding would underscore the importance of nuclear epigenetic states in shaping organelle function, with broad implications for understanding energy homeostasis and adaptation in ticks and potentially other arthropods.

Mitochondrial remodeling during the mammalian peri-implantation stage represents a critical hallmark event required for successful embryogenesis. During this period, mitochondria undergo functional and structural adaptations to meet the escalating energy demands of the developing embryo. A key component of this remodeling is the upregulation of oxidative phosphorylation (OXPHOS), which plays a pivotal role in sustaining the metabolic needs of postimplantation development. However, this metabolic shift also results in increased production of mitochondrial reactive oxygen species (ROS), increasing oxidative stress and placing mtDNA integrity at risk. Such oxidative damage, if left unchecked, can compromise mtDNA stability and ultimately jeopardize embryonic viability [69,70].

In line with this concept, our data show that BME26 tick cells treated with 5-5′-AZA, a DNA methylation inhibitor, exhibited elevated respiratory activity, indicative of enhanced mitochondrial function. However, these same cells also demonstrated a marked reduction in proliferation rates, suggesting a potential trade-off between increased mitochondrial respiration and decreased cellular growth. This observation may reflect a stress response to increased oxidative metabolism, potentially impairing cell cycle progression or activating damage-response pathways.

These findings provide new perspectives on the role of organellar epigenetics in arthropods. Given that mitochondrial function is central to embryonic development and stress responses, targeting the DNA methylation machinery might represent a novel and selective strategy for disrupting tick viability.

## 4. Materials and Methods

### 4.1. Tick Maintenance

Ticks of the species *R. microplus* (Porto Alegre strain) were obtained from a colony at the Universidade Federal do Rio Grande do Sul, Brazil in accordance with previously described procedures [58]. These ticks, which were free of *Babesia* spp., were maintained on calves in an area that lacked other tick species. Naturally detached fully engorged females. To ensure appropriate egg collection, the females were fixed in metal supports with tape and maintained in an incubator at a constant temperature of 28 °C and a humidity level of 80%. Oviposition began several days after the females were collected. During oviposition, eggs were collected daily and kept in Petri dishes within the same incubator. This collection scheme was utilized to obtain eggs over the course of 21 days, as follows: Days 6, 9, 12, 15, 18 and 21.

Embryonic fixation and DAPI staining were carried out exactly as previously described [71].

### 4.2. Genome-Wide Identification of Epigenetic Enzymes in R. microplus

We obtained the latest *R. microplus* functional annotations (BIME_Rmic_1.3) from the NCBI RefSeq *R. microplus* genome assembly BIME_Rmic_1.3–NCBI—NLM. The epigenetic enzymes from *R. microplus* were identifed after performing a BLASTP search (e-value ≤ 1 × 10^−10^, identity ≥ 25% and query coverage ≥ 50%) [72,73], using sequences of *Ixodes scapularis* epigenetic enzymes [47] against the *R. microplus* genome. *Drosophila* epigenetic enzyme homologs were also blasted against the *R. microplus* database. The epigenetic enzymes identified in *R. microplus* were double-checked by alignment with the human homologs. The conserved domain architectures were rendered with DOG (Domain Graphs, Version 1.0) [74]. Multiple sequence alignments were performed using Clustalw [75].

### 4.3. Phylogenetic Analysis

We assembled a nonredundant genome dataset from diverse arthropods, capturing key evolutionary relationships. Complete genome sequences were retrieved from GenBank, and target protein sequences from *R. microplus* were extracted and saved as .faa files. To identify orthologs, we used OrthoFinder (v3.0.1b1), selecting all orthologue groups that contained said target sequences, followed by redundancy filtering with Vsearch (v2.30.0), removing sequences with >99% identity. Multiple sequence alignment was performed using MAFFT (v7.525) with 1000 iterations, and phylogenetic trees were built using IQ-TREE2 (v2.3.6) with 1000 bootstrap replicates. Protein domain annotations were obtained using Hmmscan (HMMER v3.4) and the Pfam database, with domains filtered based on an E-value threshold of 10^−3^. Finally, data visualization was performed in R using ggplot2 (v3.5.1).

### 4.4. Molecular Docking

Amino acid sequences of target enzymes were obtained from UniProt [76]. The sequences were submitted to the AlphaFold3 server [77] with default parameters (5 cycles, template mode disabled, AMBER relaxation). The predicted structures were ranked by per-residue confidence scores (pLDDT > 80). Predicted alignment error (PAE) maps confirmed domain stability (PAE < 10 Å for intradomain residues). The models were refined via energy minimization in GROMACS 2023.1; steepest descent, 5000 steps; and protonated at pH 7.0, with charges assigned via the AMBER ff14SB force field. Grid boxes for docking were defined using residues homologous to human enzyme active sites. The human METTL3/METTL14 heterodimer, DNA methyltransferase, and histone deacetylase data were retrieved from the RCSB Protein Data Bank [78]. Polar hydrogens and Kollman charges were added using AutoDockTools (ADT v1.5.7). For docking with STM2457 (Mettl3/Mettl14 inhibitor), 5-azacytidine (5′-AZA; DNMT inhibitor), and trichostatin A (TSA; HDAC inhibitor), cocrystallized water molecules and nonessential ions were ignored. The structures were converted to PDBQT format to define atomic partial charges and rotatable bonds. Molecular docking was performed using AutoDock Vina [76], which was integrated into DockingPie 1.2.1 within PyMOL 3.1 [77]. Exhaustiveness was set to 32 for Vina, and Lamarckian genetic algorithm parameters (256 runs, 25 million energy evaluations) were used for AutoDock. The top 10 poses per ligand were clustered by root-mean-square deviation (RMSD < 2.0 Å) and ranked by binding affinity (ΔG, kcal/mol). Visual analysis of hydrogen bonding, hydrophobic interactions, and steric complementarity was performed in PyMOL.

### 4.5. Nucleic Acid Isolation and RT-PCR

Genomic DNA and total RNA were isolated from 50 mg of tick eggs at the different developmental stages (days 6 through 21 after egg laying). Genomic DNA was purified using the DNeasy Blood & Tissue Kit (Qiagen, Hilden, Germany) following the manufacturer’s instructions. Total RNA was obtained using a RiboPure Kit (Ambion, MA, USA) followed by DNase treatment (Ambion, MA, USA) and cDNA synthesis (SuperScript III First-Strand Synthesis System, Invitrogen, Waltham, MA, USA), following the manufacturer’s instructions. The PCR reactions were performed using GoTaq polymerase enzyme (Promega, Madison, WI, USA), following the manufacturer’s instructions. The PCR reactions were analyzed by 1% agarose gel electrophoresis. The *R. microplus* elongation factor 1α (ELF1α) gene was used as an endogenous control gene [79]. All oligonucleotide sequences used in RT-PCR assays are listed in the Appendix A. Quantitative RT-PCR was not performed due to low primer efficiencies, which result in high variability.

### 4.6. Dot Blot

Two hundred nanograms of genomic DNA or total RNA were spotted on a polyvinylidene fluoride membranes (PVDF, Roche, Basel, Swiss). The nucleic acids were fixed by baking the membranes at 80 °C for 1 h. The membranes were then blocked in the blocking buffer (5% skim milk in PBST) for 2 h at room temperature, and incubated with anti m^5^C or m^6^A monoclonal antibodies (Appendix A) overnight at 4 °C. After three washes with wash buffer (5% skim milk in PBST) for 10 min each, the membrane was incubated with an anti-secondary antibody (Appendix A) for 60 min at room temperature. After three washes as described above, signals were detected by chemiluminescence using the SuperSignal West Dura (Thermo fisher, Waltham, MA, USA), and an Amersham Image 600 (GE HealthCare, Chicago, IL, USA) Imaging System. Nucleic acid input controls were stained with 0.04% methylene blue. Methylene blue is a cationic dye that binds to the negatively charged phosphate backbone of nucleic acids. It allows visualization of DNA or RNA on membranes.

### 4.7. Western Blot

Protein extracts were prepared as previously described [19]. Briefly, total protein extracts from 50 mg of tick eggs at different developmental stages (Days 6 through 21 after egg hatching), were obtained by homogenization in TBS containing a protease inhibitor cocktail (Sigma, Livonia, MI, USA). Proteins were recovered from the supernatant by centrifugation at 14,000× *g*, for 15 min at 4 °C. Protein concentration was determined by the Bradford protein Assay (Bio-Rad, Hercules, CA, USA). Western blotting was carried out using the secondary antibodies (Appendix A) with a 1:5000 dilution. The primary epigenetic monoclonal antibodies (ChIP grade) used were anti-H3K9ac, anti-H3K27ac, anti-H3K27me3, anti-H3K4me3, anti-H3K9me2, and anti-H3K9me3 (Appendix A), according to the manufacturer’s instructions. For all the antibodies, a 1:4000 dilution was used. For normalization of the signals across the samples, an anti-histone H3 antibody (Appendix A) was used.

### 4.8. Statistical Analysis

Densitometry analyses were performed using ImageJ software version IJ1.46r. All the statistical analyses were performed with the GraphPad Prism statistical software package (Prism version 8.4, GraphPad Software, Inc., La Jolla, CA, USA). Statistical significance was determined using a two-tailed unpaired Student’s *t*-test. Asterisks indicate significant differences (*, *p* < 0.05; **, *p* < 0.01; ***, *p* < 0.001; ****, *p* < 0.0001; ns, not significant).

### 4.9. Cell Culture and Cell Viability Assay

The BME26 tick embryo cell line was originally obtained as previously described and maintained according to the established protocols [80]. The cells were maintained in Leibovitz L-15 medium (Sigma-Aldrich, USA), supplemented with amino acids, glucose, mineral salts, and vitamins, as previously described [80]. The medium was diluted in sterile water (3:1), followed the addition of 10% tryptose phosphate broth (Sigma Aldrich, #T8782, USA), 10% fetal calf serum (Nutricell^®^—Kota Tangerang Selatan, Indonesia, inactivated by heating), and commercial antibiotic Penicillin-Streptomycin (Gibco, #15140122, USA) was diluted in the medium (1:100), according to the manufacturer’s instructions. The cells (4 × 10^5^) were incubated on 24-well multiplates with 25, 50 or 100 μM of the epigenetic inhibitors STM2457 [59], 5-Aza-2′-deoxycytidine (Sigma-Aldrich), and Trichostatin A (Sigma-Aldrich) for 48 h. To count the cells in a Neubauer chamber, a cell suspension was placed on the grid, and the cells were counted under a light microscope. The cell viability assay was performed using the CellTiter-Glo Luminescent Cell Viability Assay (Promega).

### 4.10. Fluorescence Microscopy

A total of 4 × 10^5^ BME26 cells were cultured on glass coverslips placed in a 24-well plate. The cells were fixed with 4% paraformaldehyde for 1 h and subsequently permeabilized for 15 min in 1X PBS containing 0.15% Triton X-100. Following permeabilization, the cells were incubated in blocking buffer (1X PBS, 0.05% Tween-20, and 10% albumin) for 1 h at room temperature. The cells were then incubated overnight (16 h) at 4 °C with the appropriate primary antibodies (as listed in Appendix A), which were diluted 1:5000 in blocking buffer. After primary antibody incubation, the cells were washed three times for 10 min each in 1X PBS containing 0.05% Tween-20. Subsequently, the cells were incubated for 1 h with an Alexa Fluor 488-conjugated anti-rabbit secondary antibody (Appendix A), diluted 1:1000 in blocking buffer. Afterward, cells were washed again three times for 10 min each. Coverslips were mounted onto glass slides using 10 μL of ProLon Gold Antifade Mountant with DAPI (Thermo Fisher, USA). Immunofluorescence images were acquired using a Zeiss LSM 710 confocal microscope (Zeiss, Oberkochen, Germany).

For phalloidin staining, BME26 cells were cultured on glass coverslips in a 24-well plate. They were then fixed with 4% paraformaldehyde for 1 h and permeabilized for 15 min in 1X PBS with 0.15% Triton. To reduce the background from nonspecific staining, the cells were incubated with 1% BSA in PBS for 40 min at room temperature. For actin staining, the cells were incubated with rhodamine phalloidin reagent for 40 min at room temperature. The cells were subsequently washed with PBS three times for 10 min each. The coverslips were then mounted onto glass slides with 10 µL of ProLong Gold Antifade Mountant with DNA Stain DAPI (Thermo Fisher). Immunofluorescence images were acquired using a Zeiss LSM710 confocal microscope. For quantitative analysis of F-actin filaments we followed a previously established protocol [81].

### 4.11. Respirometry

High-resolution respirometry was performed using an Oroboros Oxygraph-2k (Oroboros Instruments, Austria) at 32 °C with a suspension of 2 × 10^6^ cells/mL in culture medium without fetal bovine serum (FBS). Basal oxygen consumption was recorded until signal stabilization. Subsequently, oligomycin (0.2 µg/mL) was added to inhibit ATP synthase and assess proton leak respiration, followed by stepwise titration of carbonyl cyanide-4-(trifluoromethoxy)phenylhydrazone (FCCP) to determine the maximal uncoupled respiration and spare respiratory capacity. Finally, antimycin A (2 µM) was added to inhibit complex III and nonmitochondrial respiration was quantified. The oxygen consumption rates were analyzed using DatLab software (version 7.4) and corrected for residual (nonmitochondrial) respiration. ATP-linked respiration was calculated as the difference between basal respiration and proton leakage, and spare respiratory capacity was calculated as the difference between maximal uncoupled and basal respiration. All the experiments were performed in three biological replicates. The data are presented as the mean ± SEM, and statistical analysis was performed using two-way ANOVA.

## 5. Conclusions

Our integrative approach revealed that epigenetic regulation is not only present but also highly dynamic and functionally significant during *R. microplus* embryogenesis.

We identified a set of conserved epigenetic enzymes, whose expression levels and enzymatic activities vary during development, suggesting precise control of gene expression and the chromatin state throughout embryogenesis.

Importantly, the dynamic interplay between H3K27 acetylation and methylation suggests a conserved Polycomb/Trithorax regulation in ticks. We also showed that m^6^A-RNA methylation peaks during early embryogenesis, coinciding with the maternal-to-zygotic transition (MZT), suggesting a role in the post-transcriptional control of gene expression.

Notably, we detected high levels of m^5^C in the nuclear DNA, and inhibition of DNA methylation led to a significant overactivation of the mitochondrial electron transport chain. These findings suggest a previously unrecognized link between DNA methylation and energy metabolism in tick embryonic cells.

Taken together, our results reveal that both nuclear and mitochondrial epigenetic regulation are critical for tick embryonic development. These mechanisms may represent promising targets for the development of novel, epigenetic-based tick control strategies.

## Figures and Tables

**Figure 1 ijms-26-09171-f001:**
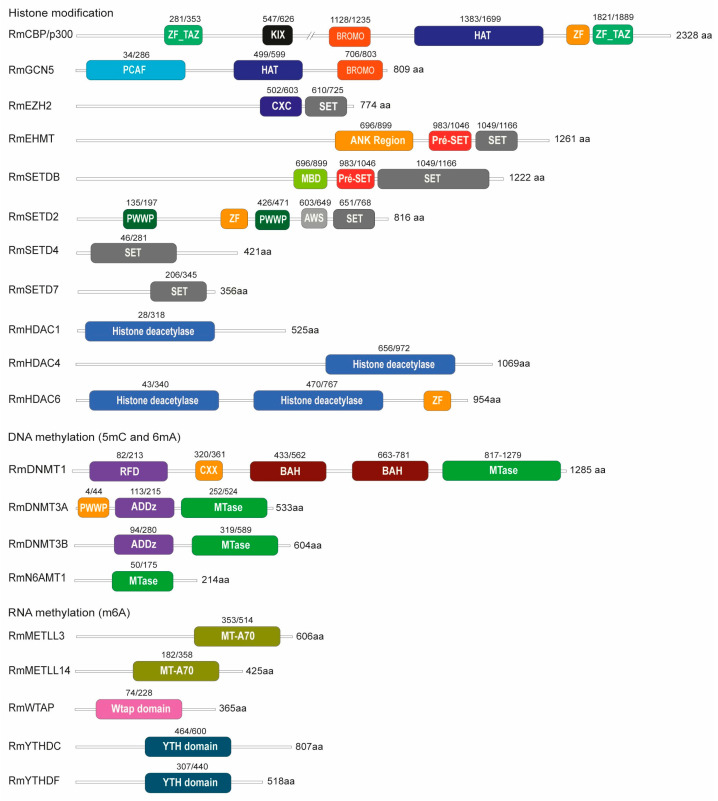
Conserved domains in epigenetic regulators of *R. microplus*. The domain architecture of the epigenetic regulators in *R. microplus* is shown, with the gene accession numbers listed in Appendix A. A comparative analysis of full-length amino acid sequences, including functional domain annotations, between *R. microplus* epigenetic regulators and their mammalian counterparts is provided in Appendix A.

**Figure 2 ijms-26-09171-f002:**
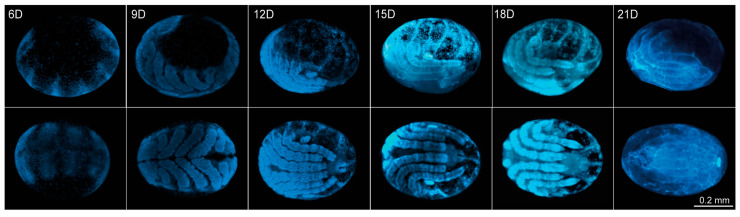
Embryonic development of *R. microplus*. Developmental stages were visualized using DAPI staining to highlight the nuclear morphology. The stages, previously described in the literature {55}, are categorized from stage 1 to stage 14, corresponding to Days 0 through 21 of embryogenesis. Owing to the lack of enough cells in eggs in the initial stages of development, we conducted experiments using eggs from Day 6 (stage 7) to Day 21 (stage14). Upper panel: Lateral view of the eggs. Bottom panel: Dorsal view of the eggs.

**Figure 3 ijms-26-09171-f003:**
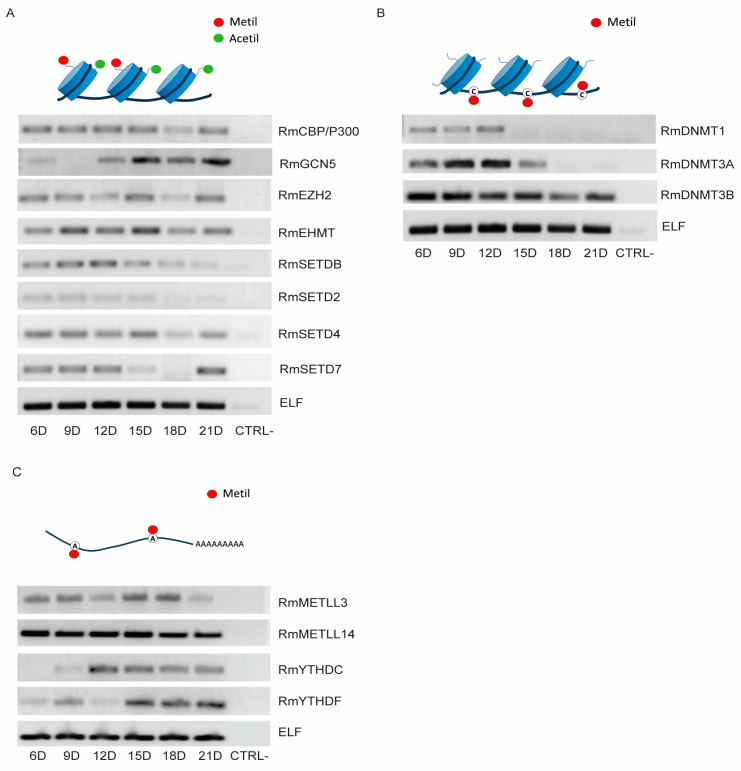
Transcriptional profile of *R. microplus* epigenetic regulators during embryonic development. Enzymes involved in histone modifications (**A**) were selected based on their well-documented roles in eukaryotic embryogenesis For m^5^C DNA methylation (**B**), the expression levels of both de novo and maintenance DNA methyltransferases were analyzed. For m6A RNA methylation (**C**), the expression levels of key writer and reader enzymes was evaluated. Elongation factor 1α (ELF) was used as the housekeeping gene for normalization. RT-PCR was performed to assess gene expression, and the amplified products were visualized by agarose gel electrophoresis. Each RT-PCR reaction was independently repeated a minimum of six times to ensure reproducibility. Negative control (CTRL-) for all RT-PCR reactions consisted of samples in which no input cDNA was added. PCR reactions were performed on tree independent biological replicates with consistent band intensity profiles.

**Figure 4 ijms-26-09171-f004:**
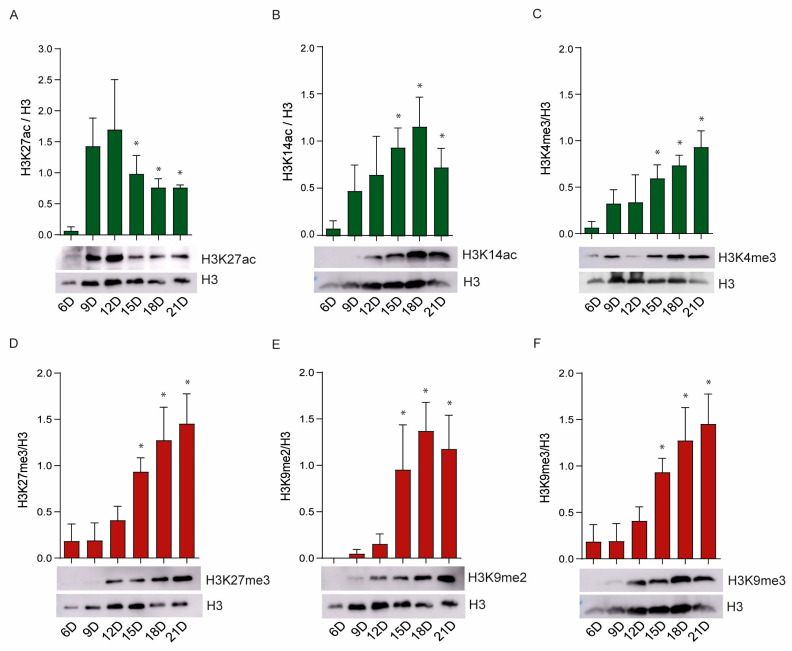
Modulation of histone modifications during tick embryogenesis. Catalytic activities of *R. microplus* histone-modifying enzymes were evaluated by Western blot analysis. Chromatin activation ((**A**–**C**), green bars) and repression ((**D**–**F**), red bars) were assessed by Western blot analysis using monoclonal antibodies. Western blotting was performed on six independent biological replicates; a representative Western blot is shown. Histone H3 was used as a loading control. The intensity of each band was quantified by densitometry analysis, and the data were plotted as a graph using ImageJ 1.54g (NIH Software). The error bars represent the standard error of the mean (SEM). Statistical significance was determined using Student’s *t*-test. *p* < 0.05 was considered significant (*).

**Figure 5 ijms-26-09171-f005:**
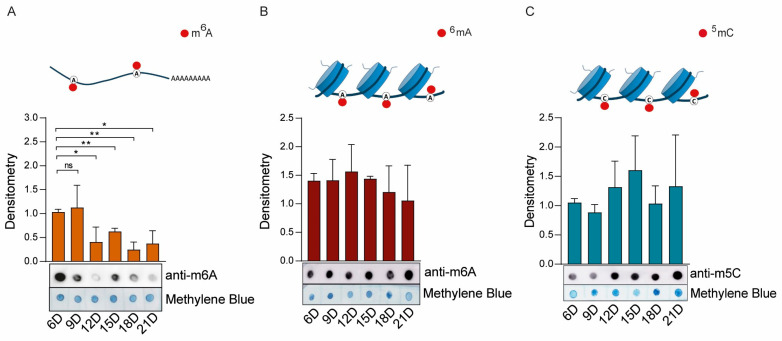
Catalytic activity of *R. microplus* m^6^A-DNA/RNA (**A**,**B**) and m^5^C-DNA (**C**) methyltransferases during embryonic development. Dot blot analyses were performed using six independent biological replicates; one representative blot is shown. Nucleic acids were stained with methylene blue and used as a loading control. Band intensities were quantified by densitometric analysis using ImageJ (NIH), and the results are presented as bar graphs. The error bars indicate the standard error of the mean (SEM). Statistical significance was assessed using Student’s *t*-test: *p* < 0.05 (*), *p* < 0.01 (**). Nonsignificant differences are indicated as “ns”.

**Figure 6 ijms-26-09171-f006:**
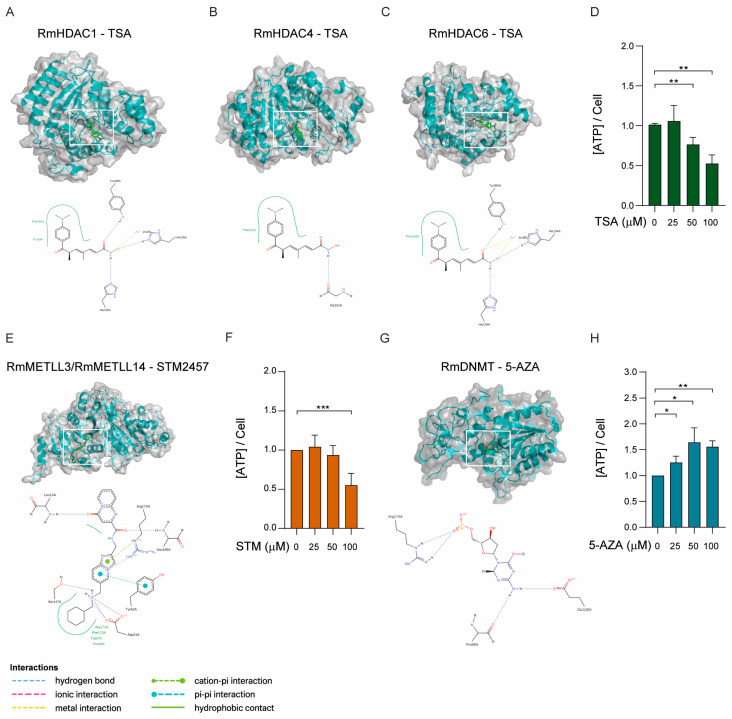
Molecular docking of epigenetic inhibitors to their respective target enzymes. (**A**–**C**) Trichostatin A (TSA; green sticks) docked into the substrate tunnel of the histone deacetylases RmHDAC1, RmHDAC4, and RmHDAC6. The hydroxamate zinc-chelating moiety coordinates to the catalytic Zn^2+^ ion, whereas the aromatic “cap” group occupies the enzyme surface groove. Hydrogen bonds to the conserved active site residues are shown as blue dashed lines. (**E**) Ribbon diagram of the METLL3/METLL14-like heterodimer (α-subunit and β-subunit), in complex with STM2457 (green sticks), docked into the conserved SAM-binding pocket. Key hydrogen bonds (dashed lines) and hydrophobic contacts are highlighted. (**G**) Docking of 5 azacytidine (5′ AZA) into the active site of DNA methyltransferase RmDNMT residues (dashed lines), and π−π stacking with a neighboring aromatic nucleotides. All panels are rendered in PyMOL; water molecules and nonessential ions were omitted for clarity. (**D**,**F**,**H**) BME26 cells were incubated with increasing concentrations of TSA, STM2457, or 5′-AZA, respectively, for 48 h, and cell viability was evaluated by measuring ATP rates. The error bars indicate the standard error of the mean (SEM). Statistical significance was assessed using Student’s *t*-test: *p* < 0.05 (*), *p* < 0.01 (**), *p* < 0.001 (***). Each ATP measuring assay was repeated at least six times. HDAC: Histone deacetylase; METLL: RNA methyltransferase; DNMT: DNA methyltransferase.

**Figure 7 ijms-26-09171-f007:**
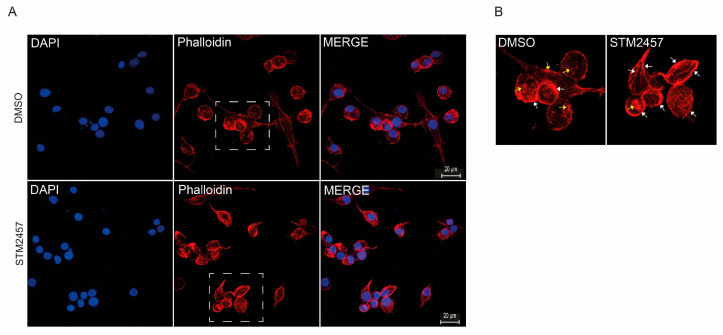
Effect of m^6^A-RNA methylation inhibition on *R. microplus* cells. (**A**) BME26 cells were treated with either 100 μM STM2457 or 5% DMSO (control) for 48 h. Nuclei were stained with DAPI, and actin filaments were visualized using phalloidin by confocal microscopy. (**B**) The region highlighted by the square in Panel (**A**) is enlarged to facilitate visualization of actin fiber organization. The yellow arrows indicate aggregates of monomeric G-actin whereas the white arrows highlight the F-actin filaments. In cells treated with STM2457 quantitative analysis revealed approximately a fivefold increase in the number of F-actin filaments compared to the control condition.

**Figure 8 ijms-26-09171-f008:**
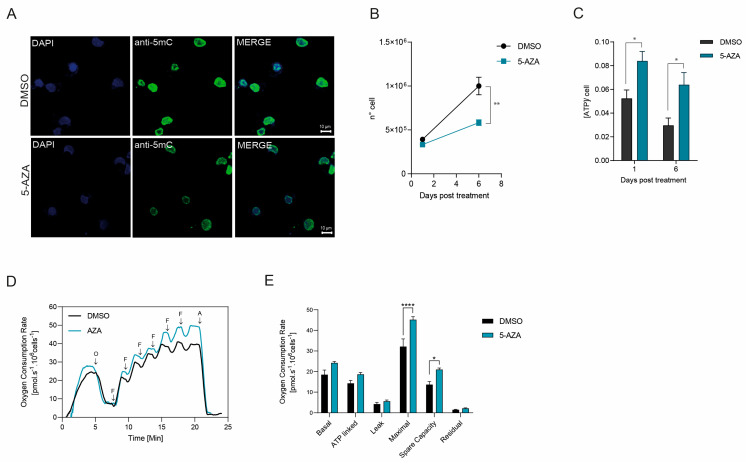
m^5^C-DNA methylation in *R. microplus*. (**A**) BME26 cells were treated with 100 μM 5-azacytidine (5′-AZA) for 48 h to suppress DNA methylation, followed by immunostaining with an anti-m^5^C monoclonal antibody (green) and nuclear counterstaining with DAPI (blue). Confocal microscopy revealed a weaker m^5^C signal in 5′-AZA-treated than in control cells confirming the effective inhibition of methylation. Scale bar: 10 μm. (**B**) The number of cells was counted 6 days after 5′-AZA treatment using a Neubauer chamber. (**C**) BME26 cells were incubated with 50 μM of 5′-AZA for six days and cell viability was evaluated by measuring ATP rates. (**D**) High-resolution respirometry of BME26 cells treated with or without AZA. Representative traces of oxygen consumption. O: oligomycin; F: FCCP; A: antimycin A. (**E**) Mitochondrial respiratory parameters derived from the respirometry data. **** *p* < 0.0001; ** *p* < 0.01; * *p* = 0.0156. N = 3, independent cell cultures. Bar graph showing the different components of mitochondrial respiration under various conditions. The parameters include basal respiration, i.e., the oxygen consumption rate (OCR) under normal, unstimulated conditions, reflecting the energetic demand of the cell at rest. ATP-linked respiration: The portion of OCR is directly couple to ATP synthesis, indicating that mitochondrial activity is dedicated to energy production. Proton leak (Leak respiration): The residual OCR is not coupled to ATP synthesis, representing protons that re-enter the mitochondrial matrix without contributing to ATP production. Maximal respiration: The maximum OCR is achieved after uncoupling mitochondrial oxidative phosphorylation, reflecting the total respiratory capacity of the cell. Spare respiratory capacity: The difference between maximal and basal respiration, indicating the ability of a cell to respond to increased energy demand or stress. Nonmitochondrial respiration (Residual): The OCR remaining after the inhibition of mitochondrial respiration, representing oxygen consumption by other cellular processes.

## Data Availability

The original contributions presented in this research are included in the article; further inquiries can be directed to the corresponding author.

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
