# Peer review of "Unraveling the Roles of Epigenetic Regulators During the Embryonic Development of *Rhipicephalus microplus"

_ijms, 2025, doi:10.3390/ijms26189171_

Round 1
Reviewer 1 Report
Comments and Suggestions for Authors
Review of Unraveling the roles of epigenetic regulators during the embryonic development of Rhipicephalus microplus by Amarante et al.
This manuscript describes the identification and baseline analysis of a subset of epigenetic regulators in R. microplus. The manuscript describes a number of well performed and important experiments and this material will be a strong basis for future work. The manuscript is generally well written, although with a number of places where clarity could be improved and some omissions of information. The writing in the introduction could be tightened up and focused. The MM has some inconsistencies between what is described in the text and the information in the MM. There are also some inconsistencies in descriptions of techniques. Many techniques are not described and simply referred to “as previously described” with a citation. This is acceptable but not overly useful. Others give a shortened protocol in addition to the citation of a standard protocol (ie 4.7), which is more useful. The results are extensive and generally well done. However several figure legends are missing information and not all supplemental figures are as described and one (S3) appears to be missing. The discussion would be greatly improved by merging the many 1-3 sentence small paragraphs into coherent paragraphs organized about distinct themes. Toning down the claims of the completeness and significance of this study would also make the study stronger. The study is important and presents a great deal of work. It does not, however, fully characterize the epigenetic regulators nor are all possible regulators identified. Aside from those which might be species specific and so not identified with this approach, the epigenetic RNA regulatory pathways are not addressed at all. Overblown claims will only serve to annoy others working in epigenetics. These comments are expanded in detail below. Despite the number of specific notes and these comments, all modifications, while necessary, are minor and this is still a strong and foundational work that definitely should be published.
Detailed notes:
Introduction: Paragraph two – the transition from economic loss due to impact on cattle health and human health (line 71) is not explained.
The rationale that this study is necessary because of acaricide resistance is contrived (paragraph 2). Paragraph 3 is a more focused and relevant justification and could be fused with a shortened paragraph 2 (the first 2 sentences of paragraph 3 are somewhat redundant). It is worth mentioning that epigenetic regulation has been also proposed in humans as a cancer therapy (ie Suraweera A, O'Byrne KJ, Richard DJ. Epigenetic drugs in cancer therapy. Cancer Metastasis Rev. 2025 Feb 26;44(1):37. doi: 10.1007/s10555-025-10253-7. PMID: 40011240; PMCID: PMC11865116 – is one of the over 1000 reviews on this topic in 2025) so the term “lower” in lower eukaryotes (line 79) is probably not necessary.
Line 99 – DNA methylation is the best studied epigenetic mechanism. It is not the most faithfully maintained – ie many genomes that use DNA methylation are completely remodelled during embryongenesis.
Intro para 6 – if mentioning the absence of DNA methylation in 2 insects, it would make as much, if not more, sense to mention majority of insects in which is it present and essential for development – any review on the topic and a short mention would address this point. DNA methylation, albeit modest, has also been found in ticks, which would be a more relevant point to bring up.
Line 108 – the “therefore” should be removed unless the role of nucleosomes in gene expression is explained.
Line 70 – typo in Anaplasma marginalev - typo
Line 76 – “viable” means living. I don’t think that is necessarily what is meant in the context of vaccines.
Line 76 “up to this date” is clunky phrasing.
Paragraph 4 of the intro could be comibined with paragraph 3. The mention of mammals (line 95) doesn’t make sense in this context as mammals have not been explicitly mentioned so far.
Intro para 9 would benefit from a segue to ease the transition from histone modifications to covalent mRNA modifications.
Last sentence of the intro should be merged into the paragraph above.
Results:
Identification of R. microplus orthologs of histone machinery contains some details that might be better left to the MM, but it is not disruptive and the inclusion of some explanatory material that could also be in the intro is useful in the results section for context.
Sup figure 1 would be improved by presented in colour. The figure legend needs to clarify to what A-Q refer.
Figure S3 does not appear to be as described in the text. Given the importance of epigenetic remodeling in the earliest stages of development, reinforced in section 2.3 which shows high levels of H mod enzymes at the earliest stages assayed shows that it would be important to include the earliest developmental stages,, even if the signal is poor. But at least these figures should be in the supplemental files.
Line 211 – italics needed.
Section 2.3 – expression profiles of various epigenetic regulators. The approach of doing RT PCR is not highly quanifiable. Why not do qPCR?
Figure 3 – what is CTRL - ? The no RT control? For which gene?
Line 234 – notably is the wrong word here. If the intent is to relate these findings to section 2.3, this idea could be expanded upon. Or the word just removed.
Section 2.4 text – the text needs to be clear on what was being detected – histone modifications or the protein expression of the histone modifiers. This is not clear from the text or figure!
Line 248 – mRNA modifications would make sense as a separate section in the results. This is a very nicely controlled set of experiments and addresses important points – genomic DNA methylation and also mRNA modification. Unless there is a limit on the number of figures that can be lncluded, the relevalnt supplemental figures would be nice in the main text.
Section 2.5 – the in silico docking study suggests that the inhibitors bind the tick enzymes but does not directly demonstrate this, as stated in Line 310. The phrasing needs to be adjusted in this statement. Similarly, the phrasing of Figure 6 legend needs to be adjusted – to indicate that this is modelled binding.
Line 317 - typo
Figure 6 legend – there appears to be a sentence in the legend from the results.
Lines 334-335 – the first sentence needs more information – it needs to state that confocal analysis was performed on treated cells and how the epigenetic modification was visualized.
Lines 335-337 – the sentence doesn’t make sense. A clear and significant what was seen with H3K27ac? Missing word?
Line 342 – as written, this states that cells were treated with both TSA and STM2457, with the implication that this was synchronous double treatment. Suggest dropping the opening clause of the sentence. It is redundant with “also” in any case.
343 – “Importantly” is without meaning in this sentence and is distracting. Further, causality is not shown in this experiment, it is merely likely that it is altered RNA methylation that impacts the cytoskeleton. The phrasing of this sentence needs to be adjusted to make it clear that causality is inferred but not demonstrated.
Discussion
Para 1 –overblown. It would be safer to say that this study improves understanding of the epigenetics of the R. microplus species rather than all ticks in general. The data on embryogenesis – ie supporting the role in development is also fairly limited. A bit of hyperbole is normal, but this paragraph could use moderation. Finally, a link between epigenetics and biological control is fairly indirect so saying that this study suggests new avenues for potential control is a stretch.
Discussion – para 2. Was this the full repertoire of epigenetic modifiers? How does one know? I suggest modifying the first sentence by removing the word “full”. What about RNA regulators?
Line 420/1 – what is “structural condensation”?
Paras 4 and 5 are background material. Suggest moving to intro or results.
Para 7 largely reiterates material in the introduction and could be cut in the discussion as it is entirely background material rather than discussing the study presented in this manuscript. The same is true of Paras 9 and, to a lesser extent, 10.
Line 487 – spacing error
Sentence 499-500 – clarity is needed. What pathways are being discussed here?
Para starting at 501 – the authors might also what to discuss the possibility of technical artifacts before positing a completely novel avenue of metabolic regulation. Also, this and the following 2 paragraphs are on the same topic – merge them.
MM
Line 552 – not a sentence
The MM indicate that qPCR was performed. But this is not apparent from Figure 3. Was qPCR done or not? And if it was done, why are ct values not provided. That would provide higher resolution than band density of agarose gels.
Line 629. Spacing error
Section 4.8 Statistical Analysis does not describe statistical analysis. This information could be placed in the relevant section. However, the name of the statistical tests performed, rather than the program used to perform it, should be provided.
Line 709 – the set of epigenetic regulators cannot be called complete. Aside from unique or yet unknown regulators, non of the RNA-mediators were studied. Suggest removing the word “complete”.
Author Response
We would like to thank both reviewers for their constructive feedback on our study. We are confident that their valuable insights have significantly improved the quality of our paper. We appreciate their time and effort, and we hope that our revised manuscript is now ready for publication in IJMS.

Reviewer 2 Report
Comments and Suggestions for Authors
need pay attention to details on M&M, result interpretation to make sure they are consistent and repeatable if needed.
Line 59-138 please add information about enzymes and encoding genes involved in epigenetic alteration from other organisms since your study mainly targeted on the epigenetic related enzymes/genes in this tick.
Line 137 change Rhipicephalus microplus to R. microplus. Do same thing for the rest of the species’ name if mentioned second time and after.
Line 208 The Fig. 3 presented here is semi-quantitative PCR. In M&M, you conducted qRT-PCR using the Power SYBR Green PCR Master Mix (Applied Biosystems) and evaluated expression level with Ct value, then you should show folder change of gene expression, not the gel image. Please explain.
comparative Ct method was used to compare mRNA abundance
Line 211 format R. microplus to italic, same for entire manuscript.
Line 246 Fig. 4., the control H3 is not consistent along with days, which means the loading is not equal or you used wrong control. How can you normalize your H3K27 expression with this unequal loading control ?
Line 263-276 (Fig. 5) and Line 618-627 (M&M), please explain why you used dot blot instead of Western blot here for m5C and m6A.
Line 331 Fig. 6. Authors conducted molecular docking analysis for relative targeted enzymes. However, there is no words in discussion. Please elaborate it.
Line 342-349 please add arrows in B-DMSO control to indicate the small aggregates of F-actin. Also difference in the number of F-actin between control (EMSO) and STM2457 treated samples (Fig. 7-B) is not very apparent.
Line 335 and 386 (Figure legend), confocal microscopy was employed to evaluate the cytotoxic effects of the inhibitors on cell morphology, but ‘confocal microscopy’ was not mentioned in M&M. Make sure your methods used in experiments are consistent with method in M&M.
Line 406 the authors jumped to a conclusion here, saying they identified a full repertoire of conserved epigenetic enzymes through this study. Are you sure you did ? how about The Ten-Eleven Translocation (TET), ubiquitination of histone H2B, or ADP-Ribosyl Transferases (ADPRTs) ?
Line 539 increased cellular growth ? not decreased cellular growth ?
Line 547-705 In entire M&M, authors didn’t mention any biological replicates for some experiment ( such as in Fluorescence Microscopy, quantitative real-time PCR, western blot), but figures presented have error bar.
Line 556 add more information about the transcriptome.
Line 561-570 please clarify how you got the orthologs from Ixodes and also other species that you used in study.
Line 563 please add accession number.
Line 583 please clarify name of amino acid sequences of target enzymes were obtained and from what species here.
Line 626-627 provide citation or brief for methylene blue staining step.
Line 709-711 how can you sure you get all conserved epigenetic enzymes identified ?
I don’t see any Table attached in review files.
For supplementary
Figure 1 please note the A-Q
Figure 2 please indicate what’s A and B are for
I didn’t see tables in supplementary folder
Comments on the Quality of English Language
Acceptable, but could be improved
Author Response

(The authors gave the same response as above.)
